# Oxygen Therapy during Exercise in Patients with Interstitial Lung Diseases

**DOI:** 10.3390/biom12050717

**Published:** 2022-05-18

**Authors:** Magda Viani, Vittoria Ventura, Francesco Bianchi, Miriana d’Alessandro, Laura Bergantini, Piersante Sestini, Elena Bargagli

**Affiliations:** 1Respiratory Disease Unit, Department of Medical Sciences, University Hospital of Siena (Azienda Ospedaliera Universitaria Senese, AOUS), Viale Bracci, 53100 Siena, Italy; viani5@student.unisi.it (M.V.); ventura@student.unisi.it (V.V.); bianchi66@student.unisi.it (F.B.); miriana.dalessand@student.unisi.it (M.d.); bergantini@student.unisi.it (L.B.); sestini@unisi.it (P.S.); 2Respiratory Disease and Lung Transplant Unit, Department of Medical Sciences, Surgery and Neurosciences, Siena University, 53100 Siena, Italy

**Keywords:** oxygen, therapy, interstitial lung diseases, pulmonary fibrosis

## Abstract

**Introduction:** ILDs are a varied group of diffuse parenchymal lung diseases associated with high morbidity and mortality. Current treatments can only slow their progression but not cure the disease. Other treatments such as oxygen therapy can also be used as support. We know very little about the effects of oxygen therapy on patients with ILDs. The aim of this study was to collect data from the literature in order to determine whether oxygen therapy can actually decrease the mortality rate or whether it is only suitable for supportive therapy for patients with ILDs. **Methods:** We reviewed the literature since 2010 on oxygen therapy during exercise in patients with ILDs. Studies that used cardio-pulmonary tests were excluded. We only reviewed those that used the 6 min walking test (6MWT) or the free walking test. We located 11 relevant articles. **Results:** All the articles except a Japanese study showed an augmentation in oxyhaemoglobin saturation during exercise when oxygen was supplied. A 2018 study called AmbOx provided important data on the effects of oxygen therapy during daily activities, showing significant improvements in quality of life. **Conclusions:** This review showed that the literature on the benefits of oxygen therapy in patients with ILDs does not provide sufficient evidence to draft specific guidelines. It was not possible to conclude whether oxygen therapy has an effect on mortality or can only play a supportive role.

## 1. Introduction

Interstitial lung diseases (ILDs) are a varied group of diffuse parenchymal lung diseases associated with high morbidity and mortality. Due to their heterogeneous clinical characteristics, ILDs are classified according to their pathology: the recent ATS/ERS classification included major fibrotic lung diseases as well as rare and unclassifiable forms of ILD [1].

One of the most important steps in the initial evaluation of an ILD patient is collecting a complete history, focused on home and working environments, hobbies and drugs taken up to that moment. The age and gender of the patient is also important because some ILDs are more common in specific age groups or show male or female prevalence. Some ILDs can present acutely, but the most common presentation is a slowly progressive onset of dyspnoea and a non-productive cough. At first, dyspnoea can occur during exercise; later it progresses to breathlessness at rest. Most symptoms are limited to the respiratory tract, but some ILDs show systemic involvement and may therefore be associated with extrapulmonary symptoms such as myalgia, arthralgia, skin lesions, eye symptoms and peripheral lymphadenopathy. Diagnosis can be obtained with a specific algorithm, including complete medical history, physical examination, lung function tests (the usual finding is restrictive deficit with altered DLCO), immunological examination and high-resolution computed tomography of the chest (HRCT) [2]. Bronchoscopy with bronchoalveolar lavage (BAL) and/or transbronchial lung biopsy can be performed under specific conditions, while more invasive biopsies can be performed if necessary for diagnosis [3,4].

A limited number of pharmacological therapies are available for ILDs. They aim to reduce inflammation, prevent fibrosis and maintain lung function parameters. For some ILDs, standard therapy includes a corticosteroid-based regimen associated with immunosuppressants such as azathioprine, mycophenolate or cyclophosphamide. For certain ILDs, avoidance of the disease-inducing exposure may improve symptoms or even lead to regression of the disease. Progressive ILDs and idiopathic pulmonary fibrosis (IPF) are the only diseases that have specific antifibrotic treatments: nintedanib and pirfenidone. Lung transplant is a therapeutic option for selected end-stage patients. Supplementary oxygen therapy is another treatment that can be of some benefit to ILD patients. Oxygen is usually prescribed to treat hypoxemia and desaturation during exertion [5]. However, limited data is available on the effectiveness of supplementary oxygen during exercise in ILD patients and there are no specific international guidelines on oxygen treatment in ILDs [6,7] (Table 1).

## 2. Ventilatory Mechanisms in ILDs

Interstitial lung diseases have low pulmonary compliance due to the reduced compliance of the lung extracellular matrix (Figure 1). This explains the restrictive ventilatory deficit characterised by low static volumes (e.g., total lung capacity), associated or otherwise with a reduction in dynamic volumes (VC, tidal volume). Early exercise intolerance can be observed in ILD patients, usually associated with rapid superficial breathing, initially during exercise, then also at rest when a patient’s condition deteriorates [8]. Ventilatory mechanics are not the only exercise-limiting cause in these patients, but there is also impaired gas exchange and circulatory limitation. Impaired gas exchange is due to the destruction of lung capillaries or thickening of the alveolar-capillary membrane, resulting in a ventilation-perfusion mismatch (V/Q mismatch), limited oxygen diffusion and low mixed venous partial pressure of oxygen (PO2). V/Q mismatch is the primary contributor to arterial hypoxaemia at rest and during exercise; however, diffusion limitation also plays a role, accounting for 19% of the alveolar-arterial oxygen pressure difference [P(A-a)O2] at rest and 40% during exercise. Patients with severe resting diffusion limitation have greater diffusion limitation during exercise. The sum of parenchymal and vascular alterations leads to a decay in respiratory function and onset of dyspnoea, first during exercise and later also at rest [9].

## 3. Oxygen Therapy in ILD

A strong correlation between desaturation during walking tests and mortality has been reported in patients with ILDs [10]. Few interventions targeted at symptom relief with robust evidence-based results have been performed for ILDs. Long-term oxygen therapy is used for people with resting hypoxaemia but it is unclear whether it is of benefit to ILD patients. The current literature does not recommend the routine use of oxygen in individuals without resting hypoxaemia. Despite this, oxygen therapy is widely used. It is recommended long-term oxygen therapy (at least 15 h/d) for patients who are hypoxic at rest, especially if there is pulmonary hypertension. Ambulatory oxygen is recommended for patients who desaturate to less than 90% during exercise if a clear benefit to exercise capacity or dyspnoea is demonstrated [11].

We found little literature on the effects of oxygen therapy in ILD patients: two studies from the 1980s and 1990s assessed the acute effect of oxygen therapy in patients with moderately severe ILD by means of bicycle endurance testing with and without oxygen therapy. Both studies showed that high flow oxygen (FiO_2_ 60%) improved endurance time (718 ± 485 vs. 680 ± 579 s, *p* = 0.01), but not dyspnoea or heart rate [12].

Although these results were important, the studies had several design limitations. Firstly, in everyday life, people rarely reach maximal effort, while on the other hand, the same FiO_2_ cannot be given to people with different severity of disease and degrees of desaturation. Subsequent studies have therefore used the 6MWT and the 6M free walking test, which assesse exercise capacity under conditions similar to everyday life. They include two small retrospective observational studies that titrated oxygen requirements to correct desaturation during an adapted 6 min walking test. Frank et al. proposed this model: if oxygen saturation fell below 90%, the test was terminated and repeated with a 2 L/min increase in oxygen flow. This titration continued until the patient no longer desaturated (SpO_2_ > 90%) or reached a flow rate of 6 L/min. Visca et al. used an approximate guide based on the experience of the Brompton Medical Centre: in patients of normal build, planning to carry their own cylinder, the required oxygen flow rate was estimated at 2 L/min for desaturations of 86–88%, increasing by 1 L for every three percentage points of desaturation. Patients desaturating <70–75% were offered flow rates >6 L/min. The estimated flow rate was increased by 25–50% in patients with a body mass index (BMI) >30, and was reduced by 0.5 L if the oxygen cylinder was not going to be carried by the subject. Correct desaturation improved walking distance, and one of these studies also indicated an improvement in Borg scores. This suggests that titration of oxygen requirements may be an important element in the implementation of ambulatory oxygen therapy [13,14].

Not all studies showed positive outcomes: Nishiyama et al. (2013) showed that oxygen supplementation of 4 L/min did not improve dyspnoea after a standardized 6 min walking test or a 6 min free walk in 20 IPF patients without resting hypoxemia. The study also showed that oxygen therapy during exercise did not improve the distance walked, leg fatigue or heart rate [15].

In one of the latest studies, Ora J et al. showed that the effect of ambulatory oxygen was significantly greater in ILD patients who desaturated during physical exertion rather than at rest. In fact, oxygen supplementation during exercise increased the exercise tolerance and reduced perception of dyspnoea in ILD patients showing exercise-induced hypoxemia. Other studies have continued this line of research [9].

## 4. Ambulatory Oxygen in Daily Life

Even though all the studies mentioned before were taken in a hospital environment, it should not be forgotten that the patients use the oxygen mostly at home during their daily life. Different efforts need different oxygen titration: Cardenosa’s study evaluated the assessment of physical activity; however, this study is unavailable as a full text as it is very recent. The design of the study is of interest, as it evaluates home oxygen monitoring in patients with ILD.

## 5. Safety Data of Ambulatory Oxygen in ILD’s Patients

Nowadays, specific safety data about ambulatory oxygen in ILD patients or specific side effects or risks are not readily available. There are, however, general contraindications to home oxygen therapy; smoking cigarettes for example increases the risk of fire. For the same reason, the oxygen supplies or the cylinders should be kept away from flames or electrical circuits. All the patients with chronic pulmonary disease in oxygen therapy should be informed of the risk of developing atelectasis, oxidative stress and peripheral vasoconstriction if the oxygen is administered in high concentrations (FiO_2_ above 50%).

## 6. Quality of Life and Psychological Implications

A randomised controlled trial (AmbOx) in the UK assessed the effects of ambulatory oxygen on health-related quality of life in patients with ILDs and isolated exertional hypoxia. In 76 subjects, they showed a statistically significant improvement in quality of life, UCSDSOBQ scores, global assessments of change in breathlessness, walking ability and activity and in the chest symptom subdomains of K-BILD scores, but not in the psychological symptom domain of K-BILD. This was the first study to focus on the daily life of patients [16,17].

The study by Ora et al., which evaluated ILD patients during the 6MWT on ambient air and during oxygen administration (mean flow 6 ± 3 L/min to maintain SpO2 nadir ≥88%), showed that if administered at the right flow, oxygen therapy in ILD patients significantly improved physical performance (in terms of distance RA: 242 ± 143 m vs. Ox: 345 ± 106 mp < 0.01) and overall dyspnoea. It also improved anxiety at rest and after exercise [16].

Another aspect of dyspnoea in ILD patients is the subjective anxiety/fear component. This is reflected in brain imaging studies, which have shown that the stimuli of dyspnoea are related to the activation of cortico-limbic areas involved in interception and nociception, and that endogenous and exogenous opioids can modulate the perception of dyspnoea [18].

## 7. Cost-Effectiveness of Oxygen Therapy

Another important chapter explored by the AmbOx study was the cost-effectiveness of ambulatory oxygen in improving the quality of life in patients with fibrotic lung diseases. The incremental cost of a unit-improvement in the total K-BILD score is estimated and the primary trial outcome was reached. The analysis considered an index, the number needed to treat, which appeared to be lower obtaining perceived improvement in breathlessness or walking ability according to the global assessment measures, than to achieve an improvement in health-related quality of life according to K-BILD.

Further steps are needed to define a preference-based utility index for the K-BILD instrument, considering preference-based measures of health as an outcome and evaluating cost per quality-adjusted life year [19].

## 8. Concluding Remarks

The literature on the benefits of oxygen therapy in patients with ILDs is insufficient to draft specific guidelines. Our aim here was to clarify whether oxygen therapy in ILD patients has a clear effect on mortality or whether it only plays an additional therapeutic role. The mechanisms underlying the improvement in patient-perceived quality of life are still unclear and no specific study has yet investigated this aspect. The next steps will be to find a way to study those mechanisms and to answer these unsolved questions.

## Figures and Tables

**Figure 1 biomolecules-12-00717-f001:**
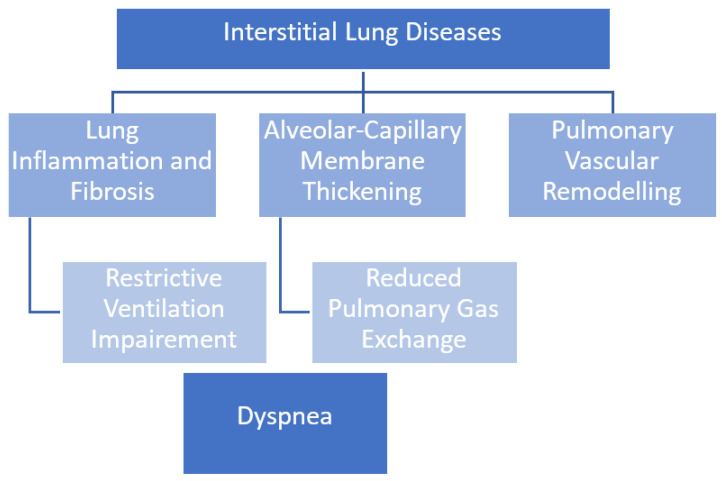
The pathogenesis of dyspnoea in interstitial lung diseases.

**Table 1 biomolecules-12-00717-t001:** Indications for Long Term Oxygen Therapy.

Continuous Oxygen
Resting PaO_2_ ≤ 55 mmHg or oxygen saturation ≤ 88%. Resting PaO_2_ 56–59 mmHg or oxygen saturation 89% in the presence of any of the following: ○ dependent oedema suggesting congestive heart failure; ○ P pulmonale on the ECG (P wave greater than 3 mm in standard leads II, III or aVF); ○ erythrocythaemia (haematocrit > 56%). Resting PaO_2_ > 59 mmHg or oxygen saturation > 89% (only with additional documentation justifying the oxygen prescription and a summary of more conservative therapy that has failed).
**Non-continuous oxygen** (oxygen flow rate and number of hours per day must be specified)
During exercise: PaO_2_ ≤ 55 mmHg or oxygen saturation ≤ 88% with low exertion; During sleep: PaO_2_ ≤ 55 mmHg or oxygen saturation ≤ 88% with associated complications, such a pulmonary hypertension, daytime somnolence and cardiac arrhythmias.

## Data Availability

Not applicable.

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
