# Peer review of "Oxygen Therapy during Exercise in Patients with Interstitial Lung Diseases"

_biomolecules, 2022, doi:10.3390/biom12050717_

Round 1

Reviewer 1 Report

This is a well-written and very interesting manuscript because, despite long-term oxygen therapy improves survival of patients with other conditions, such as chronic obstructive pulmonary disease and severe resting hypoxemia, little is known about its effect on patients with interstitial lung diseases (ILDs) during exercise. However, there are some concerns to be addressed:

  1. It would be interesting to know about safety data specific to ambulatory oxygen in ILD patients during exercise and its adverse events, side effects or risks of transporting cylinders.
  2. As physical activities in daily life are often overvalued, have you considered to include data from studies that provide a rationale for assessment of daily SpO2 monitoring and its correlation to the 6MWT in patients with ILD? (e.g. Home Oxygen Monitoring in Patients with Interstitial Lung Disease. Cardeñosa SC, et al. Ann Am Thorac Soc. 2022).
  3. In your experience, and taking into account the valuable data included in your manuscript, I would suggest to add a brief paragraph about “future directions” in this field.

Reviewer 2 Report

In this review, all the articles except a Japanese study showed an augmentation in oxyhaemoglobin saturation during exercise when oxygen was supplied. A 2018 study called AmbOx provided important data on the effects of oxygen therapy during daily activities, showing significant improvements in quality of life. This review showed that the literature on the benefits of oxygen therapy in patients with ILDs does not provide sufficient evidence to draft specific guidelines. It was not possible to conclude whether oxygen therapy has an effect on mortality or can only play a supportive role. The mechanisms underlying the improvement in patient-perceived quality of life are still unclear and no specific study has yet investigated this aspect.

I have some questions.

1) The conclusion is that the usefulness of oxygen therapy is not well established, but it is significant that such results were obtained as knowledge at this point in time.

 There are few REVIEWS summarizing oxygen therapy. This is a very important review for future findings and research.

Round 2

Reviewer 1 Report

This is a quite interesting review with very useful and practical information. The authors have assessed all the previous comments very well.

Author Response

Thanks.